# Investigation of Vermicompost Influence on Seed Germination of the Endangered Wild Rubber Species *Scorzonera tau-saghyz*

**Kenzhe-Karim Boguspaev** [1,2], **Svetlana Turasheva** [1], **Meirambek Mutalkhanov** [2,3,*],
**Zhandos Bassygarayev** [2], **Gulzira Yernazarova** [1], **Aizada Alnurova** [2] and **Balaussa Sarsenbek** [1]

[1] Department of Biotechnology, Al-Farabi Kazakh National University, 71 al-Farabi ave.,
Almaty 050040, Kazakhstan
[2] Scientific Research Institute of Biology and Biotechnology Problems, Al-Farabi Kazakh National University,
71 al-Farabi ave., Almaty 050040, Kazakhstan
[3] Department of Horticulture and Crop Science, College of Food, Agricultural and Environmental Science,
The Ohio State University, 1680 Madison Avenue, Wooster, OH 44691, USA
* Correspondence: mutalkhanov2010@gmail.com

**Abstract:** In this paper, the effect of an organic fertilizer, namely, "vermicompost tea" (VCT), on the germination of seeds of the rare wild species *Scorzonera tau-saghyz* Lipsch. et & G.G. Bosse was studied. *S. tau-saghyz* is an alternative rubber plant to *Hevea brasiliensis*, and it was widely distributed and grew well in the northwestern spur of the Tien Shan in the pre-war years (1931–1943). In recent decades, the number of wild species of *S. tau-saghyz* in natural populations has declined sharply due to climate change and the impact of anthropogenic factors. In this context, it has become necessary to restore the number of wild species. One of the critical phases in the restoration of surviving *S. tau-saghyz* populations and domestication is seed germination. The approaches that have been explored to increase seed germination, such as stratification and seed dressing with ethyl mercuric chloride, have not yielded good results. The current study covered 4 and 8 h short-term seed treatments with 1, 5 and 10% VCT. The priming of *S. tau-saghyz* seeds with 10% VCT was found to significantly increase germination from 39.0 (in the control) to 76.7% and to improve seedling vigor, mean germination time, and seedling weight. The combination of soaking the seeds in 10% VCT for 8 h and cultivating the seedlings in soil with 20% vermicompost further improved both germination and seedling growth. The vermicompost incorporation lengthened the main root, which normally accumulates rubber, and it increased its crude biomass by 1.6 times compared to that of the control.

**Keywords:** tau-saghyz; rubber-producing plant; seed; priming; vermicompost tea; vermicompost; germination



## 1. Introduction

Tau-saghyz (*Scorzonera tau-saghyz* Lipsch. & G.G. Bosse) translates from Kazakh as mountain gum ("tau"—mountain; "saghyz"—gum), and this plant has dry roots containing up to 40% natural high-quality rubber. The plant was mentioned for the first time in 1929. Research at that time revealed that this plant grows only in Kazakhstan in the highlands of Syr-Darya Karatau. It was also discovered that the natural thickets of tau-saghyz cannot supply enough natural rubber for the local industry. Therefore, the subject of the development of tau-saghyz and its introduction into cultivation emerged [1,2]. *S. tau-saghyz* is a semi–shrub with short, thick subterranean stems (caudexes) terminating in a rosette of linear leaves, and it is distinguished by its sluggish growth and development. The rubber/latex in tau-saghyz, like that in other known rubber-containing plants (*Hevea brasiliensis*, *Taraxacum kok-saghyz*, etc.), is synthesized in specialized milky cells and serves as a barrier function by protecting plant-damaged areas from insect invasion and/or preventing feeding by the sticking (pruning) of the oral apparatus of insect pests [3,4].

The method of rubber extraction with hexane was utilized in our previous study to examine the quality of tau-saghyz rubber. The extracted rubber structure was determined using 1D NMR spectra, namely, $^1$H, $^{13}$C, and DEPT, and two-dimensional homonuclear correlation COSY $^1$H-$^1$H, heteronuclear correlation HMQC $^1$H-$^{13}$C, and HMBC $^1$H-$^{13}$C. These analyses revealed the sample's polyisoprenoid composition and perfect concordance with the natural rubber described in the literature [5]. The impact of low temperature on the germination of the tau-saghyz seeds in a culture was also proved to be effective. The chosen seeds soaked in water were kept at a temperature of +20 °C for 12 days. Germination increased from 54.0 to 73.6% when they were placed in a thermostat (temperature +25 °C). Simultaneously, germination was completed within 4 days, but germination in the control required 12 days. However, when sowing wet seeds in the soil on a field scale, their pre-treatment with low temperatures proved to be technically challenging. In March, the seeds in bags were placed on the snow stored on the fields, and then they were manually sowed. This process was also challenging.

We leveraged our experience of pre-treatment with "vermicompost tea" (VCT) in trials on maize (*Zea mays*) seed germination in the quest to identify affordable and effective techniques for treating (priming) tau-sagyz seeds before sowing, [6,7]. Our modified technology of mixing pre-fermented cow manure with *Eisenia fetida* worms was employed in experiments with vermicompost. The pathogen antagonists found in the vermicompost included *Trichoderma viride*, *Trichoderma hamatum*, *Pseudomonas aeruginosa*, *Xanthomonas maltophilia*, *Enterobacter cloacae*, and *Bacillus subtilis* [8,9]. Even with all of the advanced agrotechnical measures available at the time, the tau-sagyz yield was inconsistent with that in the plantations established in 1933–1935. A low seed germination, pest infestation (root nematode, insect "tau-sagyzka"), a high polymorphism, seed collection issues, and the advent of synthetic rubber all contributed to the decision to minimize the area used for *tau-saghyz* cultivation. Kok-saghyz (*Taraxacum kok-saghyz*) was left in the USSR for cultivation. Since then, no one in the world has addressed the challenge of cultivating tau-saghyz for over 80 years.

This study aimed to develop techniques to increase the germination, growth, and development of tau-saghyz.

## 2. Materials and Methods

### 2.1. Plant Material

The seeds of the wild species *Scorzonera tau-saghyz* Lipsch.et & G.G. Bosse were obtained from 3–5-year-old plants grown on the territory of the Karatau State Nature Reserve in the upper valleys of the Karatau Mountains (coordinates: 43°33′13″ N, 68°53′52″ E, 816 m above sea level) (Figure 1A–D).

The seeds were manually collected by cutting the basket during the last 4–5 stages of growth, when the basket is fully open and shaped like a ball (Figure 1B). The collected seeds were dried in the shade on a grid with 1 mm-thick cells on the first day and then in the sun with regular stirring. The trials used undamaged seeds that were devoid of down (Figure 2).

### 2.2. Vermicompost and "Vermicompost Tea" Preparation

Two types of vermicompost were used: dry and liquid vermicompost. The vermicompost was produced using a modified self-moving heap method developed in our laboratory, in which compost worms, namely, *Eisenia fetida*, from cattle manure were employed in open areas over the summer [10]. The "vermicompost tea" (VCT) was made using chlorine-free settled water and dry, sieved vermicompost (*v/v*). A CX-0088 compressor was used to aerate the aqueous suspension. The preparations were conducted with a constant aeration of the suspension for 1–3 days. The resultant suspension was diluted several times with distilled water to acquire varied concentrations of VCT.

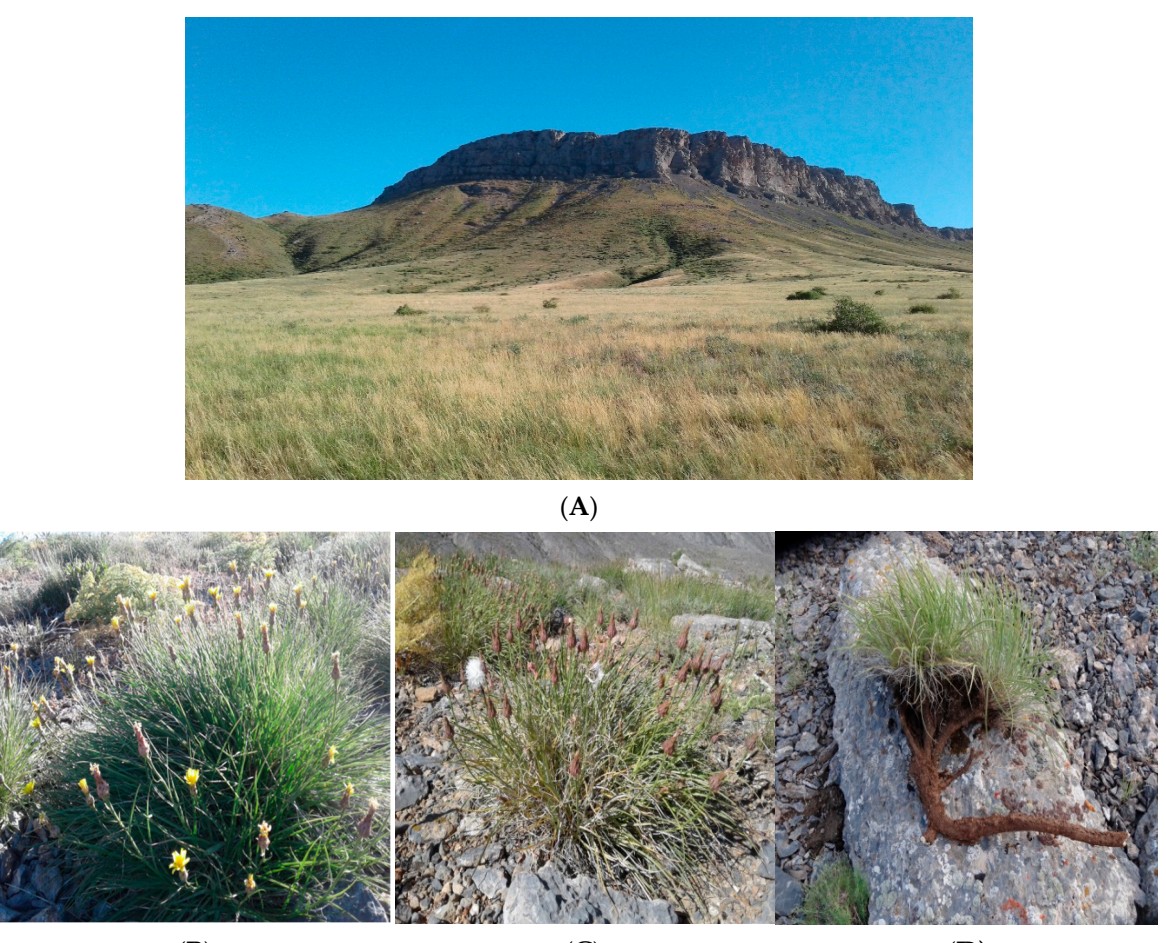

**Figure 1.** The rare, endangered rubber species *Scorzonera tau-saghyz* Lipsch. et G.G. Bosse. (**A**) Karatau ridge—a native habitat of the wild species *S. tau-saghyz*; (**B**) wild plants in the flowering stage; (**C**) stage of plant development suitable for seed collection (anthodium in the last stage of development); (**D**) root of tau-saghyz.

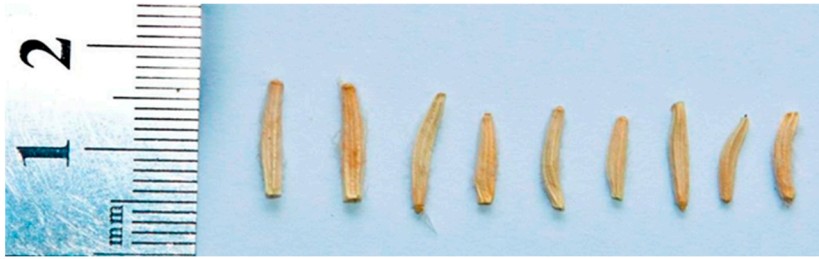

**Figure 2.** Mature *S. tau-saghyz* seeds.

*2.3. Seed priming and Seedling Production*

The dry seeds were soaked in 1, 5, and 10% (*v/v*) VCT for 4 and 8 h at room temperature. The seeds steeped in distilled water served as controls. The seeds were then placed in Petri dishes lined with a moist filter paper. The experiments were conducted in three replicates, with 100 seeds in each variant, and the seeds were watered every three days throughout the experiment. The seeds were germinated in a greenhouse under the following conditions: a photoperiod of 16 h and a temperature of $25 \pm 2\ °C$. To assess the percentage of germination and seedling growth, measurements were performed every three days.

*2.4. Cultivation of S. tau-saghyz in Standard Vessels with 20% Vermicompost-Enriched Soil*

Stuewe & Sons, Inc., Tangent, OR, USA, equipment was utilized for the mass production of seedlings in the laboratory. The standard vessels had a capacity of 200 mL. The illumination intensity was 5000 lux, with a 16 h photoperiod. The plants were irrigated twice a week.

For a plant growth analysis, the following parameters were measured every 6 days for 6 months: shoot length (mm), the number of leaves, taproot length (mm), the number and length of the lateral and axillary roots, and total biomass (above and below ground fresh biomass, dry biomass (mg)). The shoots and roots were dried to a constant weight in a thermostat at 75 °C in order to determine their dry weight (g). An average value with a standard deviation was calculated to assess the parametric data in each experiment. Student's criterion for independent samples was used for the statistical processing of the experimental data on the influence of VCT on seed germination and the data on groups of vermicompost and seedling growth characteristics [11].

## 3. Results

*3.1. Seed Pre-Treatment and Seedling Procurement*

Priming is a simple, low-cost, and highly efficient method of improving plant stress tolerance, ensuring uniform and rapid seed germination, and encouraging root system development. Priming was applied in our experiments using VCT, a seed treatment that includes phytohormones and biologically active compounds. The results of the experiments are shown in Table 1.

**Table 1.** Wild *S. tau-saghyz* seed germination following 4 and 8 h seed pre-treatment with 1, 5, and 10% VCT.

| VCT Concentration, % | Seed Germination Rate (%) after Pre-Treatment with VCT * | |
| --- | --- | --- |
| | **4 h** | **8 h** |
| control (without VCT priming) | $31.7 \pm 4.66$ | $40.0 \pm 4.92$ |
| 1 | $43.0 \pm 3.78$ | $45.0 \pm 3.80$ |
| 5 | $50.0 \pm 5.01$ | $60.0 \pm 4.92$ |
| 10 | $68.3 \pm 4.68$ | $76.7 \pm 4.25$ |

* Note: the lengths of seedlings in the experiment and control after 4 and 8 h of exposure differ ($p < 0.05$).

Figure 3 depicts the results of a typical experiment on the germination of tau-saghyz seeds when treated with different VCT concentrations (control—without treatment).

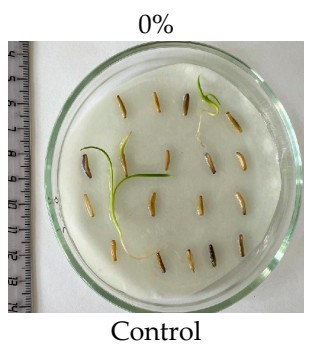 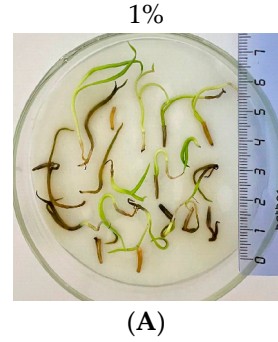 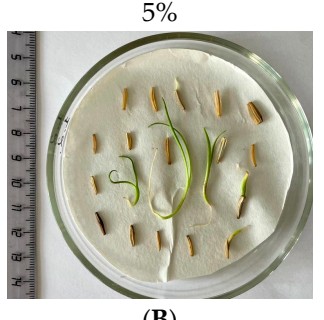 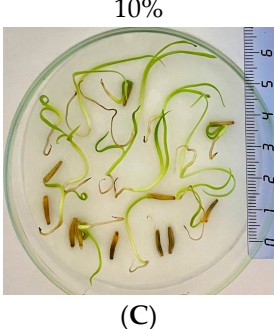

**Figure 3.** Seed germination of *S. tau-saghyz* seeds following a 4 h pre-treatment with various VCT concentrations (12th day): (**Control**) without VCT priming; (**A**) 1% VCT; (**B**) 5% VCT; (**C**) 10% VCT.

An analysis of the obtained data showed that priming the dry tau-saghyz seeds with VCT at all used concentrations resulted in an increase in the number of seedlings

compared to the control. Priming with 5 and 10% VCT significantly increased the number of germinated seeds irrespective of the exposure time. However, the treatment of the dry seeds with a 10% solution of VCT for 8 h was the most effective. Here, the percentage of germinated seeds was 68.3 ± 4.68% for a 4 h exposure time and 76.7 ± 4.25% for an 8 h exposure time, outperforming the control by 2.1- and 2.4-fold for 4 and 8 h exposure times, respectively.

### 3.2. Vermicompost Influence on Tau-Saghyz Seedling Growth

Vermicompost, a well-known organic fertilizer produced in our laboratory from cattle manure using the compost worms *Eisenia fetida*, was employed to improve tau-saghyz growing methods. Vermicompost is used to enrich the soil biota with microorganisms that produce plant growth and development compounds, such as phytohormones. A soil mixture of two components—vermiculite and washed river sand—was utilized in a 1:1 (v.v) ratio without the addition of soil or mineral fertilizer in order to evaluate the effect of the vermicompost.

Consequently, it was discovered that adding 20% vermicompost by volume to the soil boosted plant growth parameters substantially. Thus, the height of the *S. tau-saghyz* seedlings was 1.2 times greater than that of the control in the first week of growth, and on the 16th day of cultivation, the plants reached a height of 73.6 mm, which was 1.36 times greater than that of the control. The beneficial effect of utilizing 20% vermicompost was established with a 95% certainty (Figures 4 and 5).

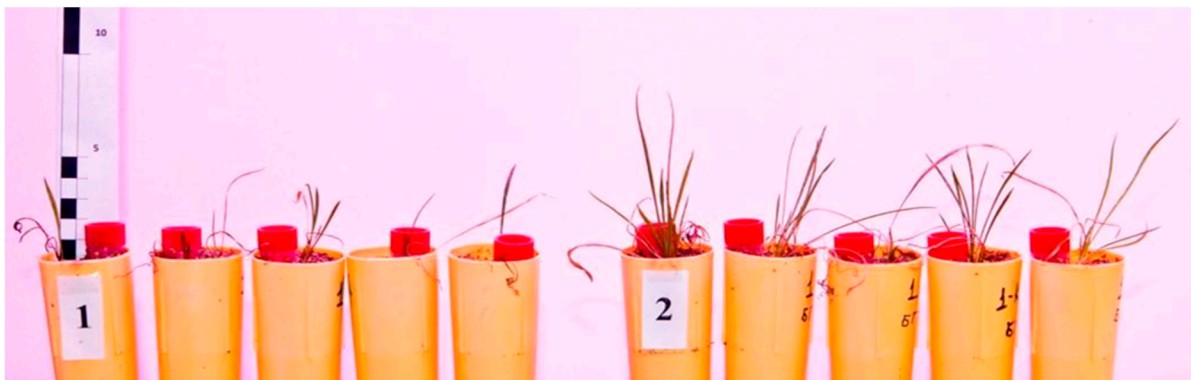

**Figure 4.** Seedlings of *S. tau-saghyz* without vermicompost (**1**) and with 20% vermicompost (**2**) after 172 days of growth.

The average values of the seedling height and the number of leaves generated during the experiment were compared to provide a more reliable picture of the impact of the vermicompost. Figure 5 depicts the results. The average values of the plant height and leaf quantity were about 1.5 times greater than those in the control. The average plant heights in the experiment and the control were 61.4 ± 3.20 and 40.1 ± 2.40 mm, respectively, whereas the leaf quantities were 9.00 ± 0.50 and 6.13 ± 0.40, respectively (Figure 6).

The fresh weight and dry weight (FW and DW, respectively) of the vegetative organs (leaves and stems), roots, and the entire plant were measured to assess the effect of the vermicompost on the tau-saghyz seedlings.

The crude weight of the aerial part of *S. tau-saghyz* averaged 83.3 ± 6.3 mg in the control, while it was three times greater in the experiment, averaging at 252 ± 18.1 mg. In the control, the average plant FW was 220 mg; in the experiment, it was 476 mg, which is 2.19 times greater (Figure 6). In order to test the hypothesis that the more developed the leaf, the greater the root system, the FW of the Tau-saghyz leaves and roots in the control (1) and experiment (2) were compared (Figure 7).

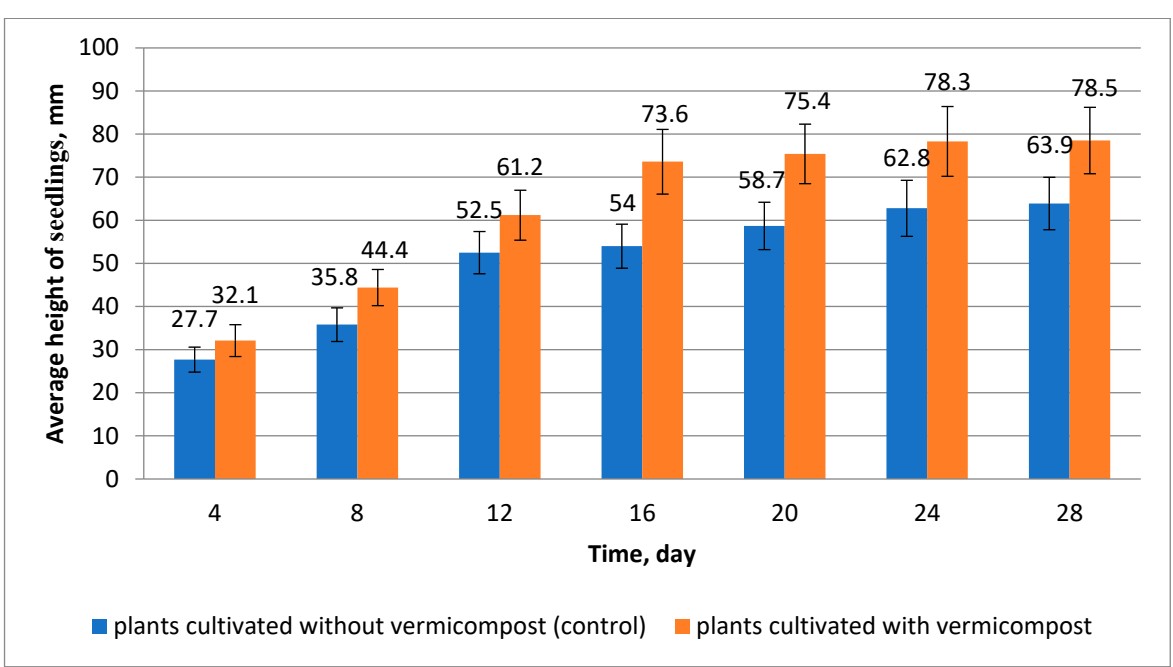

**Figure 5.** Vermicompost influence on *S. tau-saghyz* seedling height.

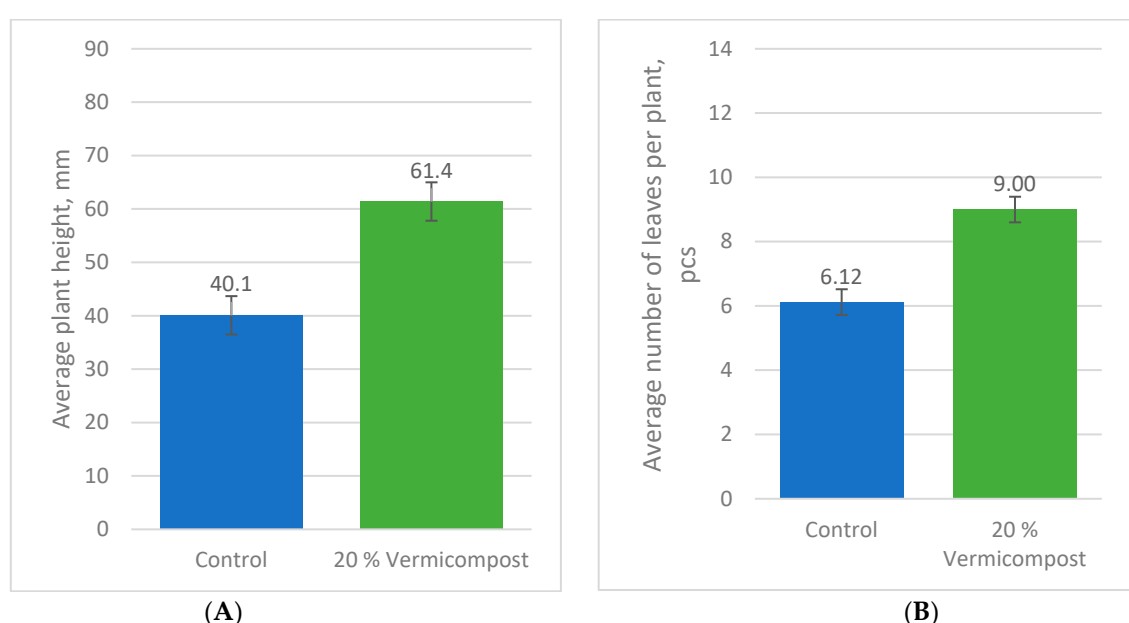

**Figure 6.** Vermicompost's impact on S. tau-saghyz height (**A**) and leaf quantity (**B**).

1. In the control, the average FWs of the leaves and roots corresponded to 83.3 and 136.6 mg, respectively, and the ratio of the average weight of the raw roots to the weight of the raw leaves was 1.63. There was thus a clear correlation between the FW of the Tau-saghyz roots and leaves in the control, allowing for an estimation of the root system growth based on leaf development. A coefficient of 0.61 was obtained when assessing the link between the average FWs of the leaves and roots, demonstrating a direct correlation; i.e., when the leaves' weight increases, the root weight grows accordingly.

2. In the experiment, the average FWs of the roots and leaves were 223 and 252 mg, respectively, and the ratio of the FWs of the roots to leaves was 0.88; this was much lower than that in the control (less than 1), indicating that there was no direct correlation in this case. The ratio of the leaves to the roots FW was 1.13; this is above 1,

indicating the absence of a direct correlation. Therefore, it can only be concluded that vermicompost has a greater influence on leaf development.

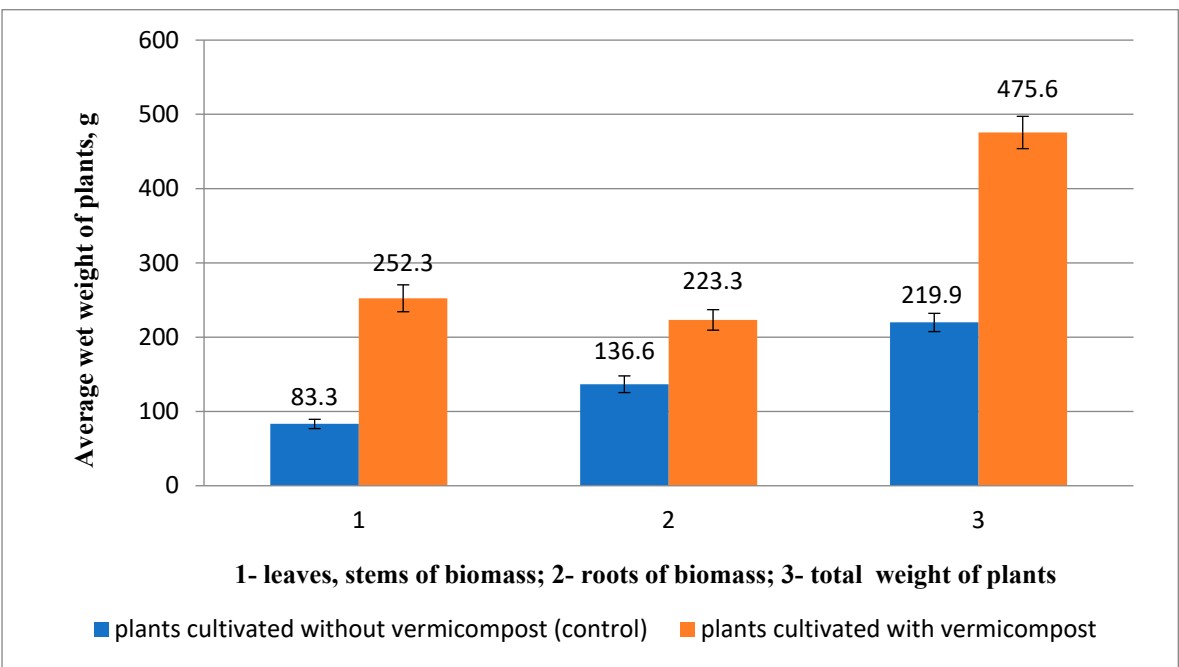

**Figure 7.** Vermicompost influence on the S. tau-saghyz fresh weight.

To determine plant DW, the suspended samples of the fresh biomass of the experimental and control plants were dried to a constant weight in a thermostat at 75 °C. The DW of the plant aerial part averaged at 16.3 ± 1.40 mg in the control, whereas in the experiment, it was more than twice as high, averaging at 38.0 ± 3.10 mg. However, there was a slight difference when comparing the root DWs between the experiment and the control, being equal to 29.3 and 28.0 mg, respectively.

The following peculiarities were discovered when processing the data on the ratio of the DWs of the roots to leaves:

1. In the control, the ratio of the DWs of the roots (28.0 mg) to leaves (16.3 mg) was 1.71.
2. In the experiment, the ratio of the DWs of the leaves (16.3 mg) to roots (28.0 mg) was 1.17, being more than 1. There was no positive correlation detected in the experiment, which was presumably due to the vermicompost primarily influencing the leaf biomass growth and subsequently allowing for the root system to also expand (Figure 8).

Thus, adding 20% vermicompost to the soil improved the growth and development of the Tau-sagyz plants. The seedlings cultivated on the soil enriched with vermicompost developed significantly better than the control seedlings. The vermicompost stimulated the development of the leaves and stems, but only the FW increased in the case of the roots. This finding does not contradict the widely held belief that roots collect nutrients from the soil and transport them to the leaves. Only until the biomass of the leaves increased as a consequence of photosynthetic activity and rubber biosynthesis could organic substances penetrate the roots, increasing their size and, therefore, the DW (Figure 8).

The morphology of the *S. tau-saghyz* root system was investigated to help elucidate this issue. The data analysis revealed that there were no significant differences between the experimental and control plants in terms of characteristics such as the length of the main root and the number of subsidiary roots (Figure 9) (Tables 2 and 3).

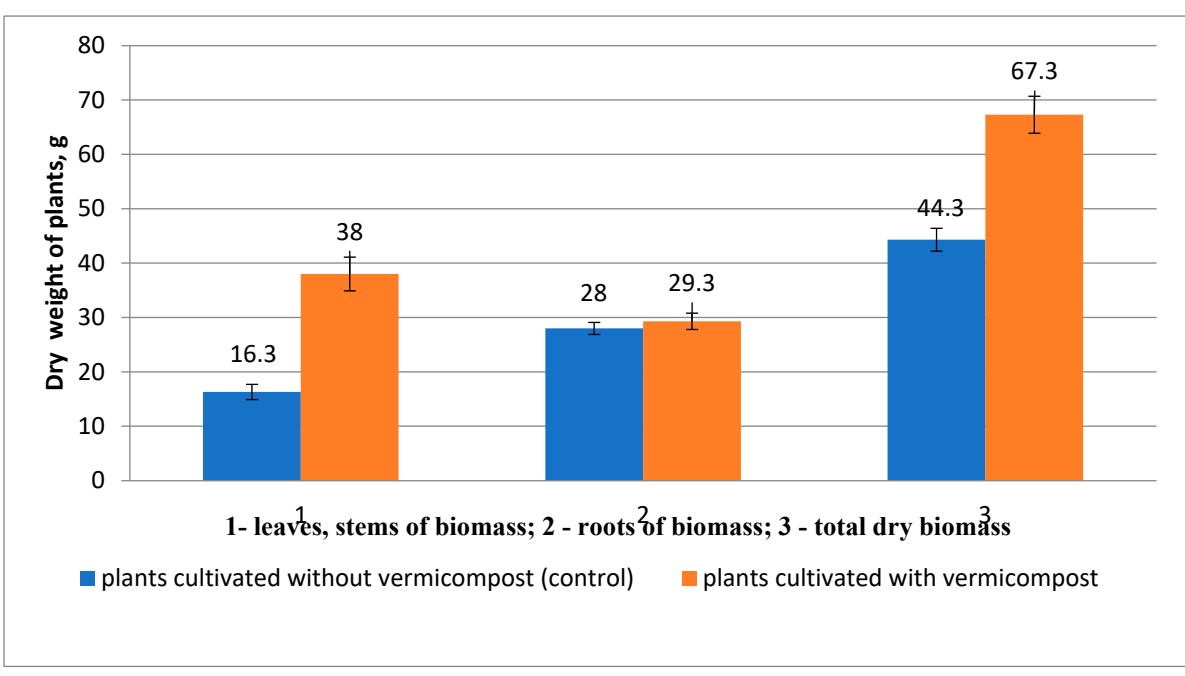

**Figure 8.** Vermicompost influence on the S. tau-saghyz dry weight.

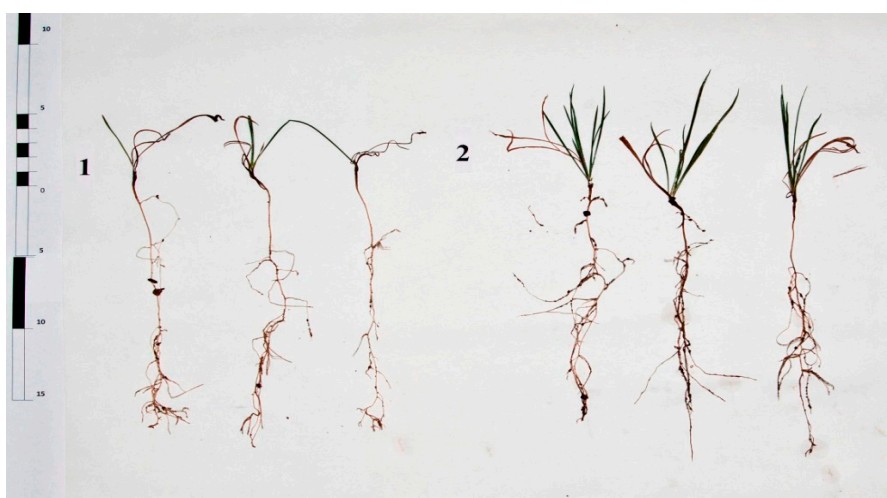

**Figure 9.** The root system of *S. tau-saghyz* seedlings ((**1**): control; (**2**): experiment).

**Table 2.** Influence of 20% vermicompost on S. tau-saghyz root system development (on 172nd day).

| Root System Parameters | | Control | Experiment |
|---|---|---|---|
| Main root length, mm | | 196 ± 9.90 | 200 ± 11.5 |
| Amount of lateral roots sprouting from the main root, pcs | | 1.30 ± 0.10 | 2.70 ± 0.20 |
| Length of lateral roots sprouting from the main root, mm | 1st order | 0.10 ± 0.01 | 0.20 ± 0.01 |
| | 2nd order | - | - |

Sample values are significantly different ($p < 0.05$).

A comprehensive investigation of the morphology of the *S. tau-saghyz* root system, namely, the main root system, revealed that the lateral roots grew better in the experiment (20% vermicompost) than in the control. Thus, in the experiment, the number of lateral roots sprouting from the main root averaged at 2.70 ± 0.20, representing a statistically significant two-fold increase over the control (Table 2). The length of the first-order lateral

roots of the plants grown in the 20% vermicompost-enriched soil was 4 times more than that of the control, averaging at 15.0 ± 0.10 and 3.50 ± 0.20 mm, respectively (Table 3).

**Table 3.** Influence of 20% vermicompost on S. tau-saghyz axillary root development.

| Root System Parameters | | Control | Experiment |
|---|---|---|---|
| Number of axillary roots, pcs | | 18.7 ± 1.20 | 19.0 ± 1.40 |
| Length of axillary roots, mm | | 83.3 ± 3.10 | 96.7 ± 4.20 |
| Number of lateral roots sprouting from the axillary root, pcs | | 1.30 ± 0.10 | 1.30 ± 0.10 |
| Length of lateral roots sprouting from axillary roots, mm | 1st order | 3.50 ± 0.20 | 15.0 ± 0.10 |
| | 2nd order | - | - |

Sample values are significantly different ($p < 0.05$).

The *S. tau-saghyz* seedlings' growth and development were enhanced by the addition of the 20% vermicompost. In the experiment, the average height and number of leaves were 1.5 times higher than those of the control. The DW of the aerial part of the *S. tau-saghyz* seedlings was more than twice as high as that of the control. Most of the analyzed characteristics of the *S. tau-saghyz* plants' main and axillary roots did not change considerably. However, in the experiment, the number of lateral roots sprouting from the main root and the length of the lateral roots sprouting from the axillary roots (first-order) were 2 and 4 times higher than those of the control, respectively. The DW of the entire plants was over 1.5 times greater than that of the control.

The growth indicators of the examined *S. tau-saghyz* plants cultivated in the 20% vermicompost-enriched soil for 172 days show that a single application has a long-term favorable effect for the full growing season.

## 4. Discussion

Nowadays, there are no more reliable numbers, but natural Tau-saghyz thickets cover rather large territories within 10–12 thousand km$^2$ in the Karatau Mountains, corresponding to 6–8 million plants. The yield of such a vast plant number is not enough to fulfil the yearly demand for natural rubber, which is limited in Tau-saghyz thickets. Therefore, it is necessary to restart the introduction of native rubber plants, beginning with the establishment of plantations that will allow for sufficient amounts of natural rubber to be collected.

Tau-saghyz reproduces naturally via seed and vegetative reproduction. Vegetative reproduction occurs as a result of the stem shoots on the root system. This reproductive mechanism is the most prevalent in natural Tau-saghyz thickets, resulting in the extensive expanses of Tau-saghyz thickets. Cuttings with both caudex and root segments do not yield satisfactory results on a commercial scale; however, seed propagation does.

Seed reproduction in Tau-saghyz habitats is a very rare phenomenon. No shoots were detected on the northern and southern slopes of the Karatau Mountains (the Jelagan-ata tract), according to our observations. A single shoot was discovered only on the southeastern slope. Calculations of the amount of potential yearly Tau-saghyz development via seed propagation, however, have indicated that the thickets produce up to 10 tons of seeds annually (1936–1938), i.e., around 1.5 billion seeds, assuming that one-third of the seedlings are 500 million seeds. Even if half of the plants would die (due to pests and diseases), the yearly increase should be 250 million plants, although in nature, only 7–8 million plants grow. Based on Kultiasov's research and our observations, it can be concluded that the unfavorable conditions for seed germination and Tau-saghyz growth in the Karatau Mountains are limiting factors for plant expansion. The first studies on the germination of Tau-saghyz seeds were conducted to determine the effect of physical factors, light, and temperature while considering the stage of seed development when collected in the wild. The optimal temperature for seed germination was discovered to

be 20–25 °C [12]. Seeds should be collected during the fourth and fifth developmental stages for sowing. Low temperatures (2 °C) were also tested in a previous study for their influence on seed germination. Fresh seeds were refrigerated for 8–10 days before being placed on a thermostat set to 25 °C. As a result, the following conclusions were drawn: (a) low temperatures during Tau-saghyz seed sowing positively affect seed germination, dramatically increasing germination energy; (b) as germination increases, the percentage of seed death from fungal diseases decreases; and (c) Tau-saghyz seeds must be free from pubescence to be sown [13]. These factors are significant and should be considered when choosing the period for sowing Tau-saghyz in field studies. Moisture absorption by seeds is recognized as an initial mechanism of seed germination, serving as the foundation for priming, a common method of enhancing seed germination that involves pre-treatment with biologically active compounds [14]. VCT, which contains soluble macro- and micro-elements, as well as bacteria in various concentrations (1, 5, and 10%) (v.v), was utilized in our trials to stimulate the germination of Tau-saghyz seeds. It should be noted that the processing time affects seed germination and aids in the removal of dormant seeds. Since *S. tau-saghyz* seeds have a dense shell, a preliminary 8 h treatment is required to bring them out of dormancy. The percentage of germination was relatively high with an 8 h treatment of 10% VCT compared to a similar exposure of 5% VCT. Most seeds are pecked after 2–3 days after soaking. Thus, 10% VCT is a cheap and efficient liquid biological product that can stimulate seed germination in wild species *S. tau-saghyz*.

Growing the seedlings in soil enriched with 20% vermicompost increased the examined parameters. It was discovered that, after 16 days of cultivation, the height of the Tau-saghyz seedlings increased by an average of 46.4 mm, whereas that of the control increased by 36.2 mm. The quantity of photosynthetic green leaves increased after the first week of cultivation, peaking on the 16th day. Similar results were observed in a previous study on spinach and squash production using 5 and 10% vermicompost (from cattle manure) for 35 days. The beneficial impact of vermicompost was proven by an increase in the number of leaves, fresh, and dry weight of the leaves and stems, leaf area, and dry weight of spinach roots [15].

The quantity and quality of natural rubber define the economic value of tau-saghyz. Accordingly, determining the parameters related to increased rubber content is a foundation for the introduction of tau-saghyz into the culture and the construction of an economically viable rubber culture. According to Kultiasov, "the more developed the leaf mass, the greater the root mass should match to it", implying that the leaf mass is an indirect criterion for root mass output. Hence, the leaf may be used to estimate the accumulation of rubber in the root mass. Since photosynthesis is the leaf's primary function, the amount of photosynthetic pigments was chosen as an indicator in our previous experiments, possibly correlating with the rubber accumulation in roots. Consequently, in all of the examined groups, a correlation analysis indicated a very weak link between the pigment amount in the leaves and the share of rubber in the roots. Thus, the amount of photosynthetic pigments does not correspond to the rubber accumulation in the roots [16]. Our preliminary studies of Tau-saghyz wild thickets revealed that the rubber content is differently distributed over the whole length of the root. The rubber content is the highest near the beginning of the root and progressively diminishes along its length. Rubber is the most abundant (synthesized in leaves) following the initial blooming and fruiting of plants and continues to accumulate throughout the plant's life. The root may grow up to 1 m in length. However, no mechanism has been discovered to precisely assess Tau-saghyz age, life expectancy, or capacity to accumulate rubber. Meanwhile, root length has no effect on the overall amount of rubber; according to our unpublished data, the root weight and rubber amount have no direct relationship. All of these uncertainties need to be further researched in future studies.

## 5. Conclusions

Under natural conditions, wild *S. tau-saghyz* plants reproduce mostly via the vegetative mechanism (the branching of roots and caudexes), with seed reproduction being

insignificant. Seedlings are frequently killed by the spring temperature drop, pecked by birds, and attacked by pathogenic microorganisms and insects. However, the seeds are still utilized to establish plantations, and the appropriate conditions for their growth are provided because the plants grown in this manner outperform wild species in terms of viability and rubber accumulation. Current research has shown that employing a vermicompost rich in mineral and organic compounds can alleviate the problem of low seed germination and enhance viability and seedling growth. The investigated method of treating the seeds with 10% "vermicompost tea" for 8 h, which increases the percentage of seed germination of wild *S. tau-saghyz*, and planting the *S. tau-saghyz* seedlings in 20% vermicompost-enriched soil can be recommended in measures for the restoration of natural populations of *S. tau-saghyz* and commercial rubber production.

**Author Contributions:** Conceptualization, K.-K.B.; methodology, K.-K.B. and S.T.; software, M.M.; validation, K.-K.B. and S.T.; formal analysis, K.-K.B., S.T. and M.M.; investigation, G.Y., A.A. and B.S.; resources, Z.B.; data curation, K.-K.B. and M.M.; writing—original draft preparation, K.-K.B. and M.M.; writing—review and editing, K.-K.B., M.M. and S.T.; visualization, K.-K.B. and M.M.; supervision, K.-K.B.; project administration, K.-K.B. and M.M.; funding acquisition, K.-K.B. All authors have read and agreed to the published version of the manuscript.

**Funding:** This research was funded by Ministry of Science and Higher Education of the Republic of Kazakhstan, Grant number. 08053131.

**Institutional Review Board Statement:** Not applicable.

**Informed Consent Statement:** Not applicable.

**Data Availability Statement:** The data presented in this study are available on request from the corresponding author. The data are not publicly available due to being located in laboratory journals.

**Acknowledgments:** An employee of the Karatau State Nature Reserve.

**Conflicts of Interest:** The authors declare no conflict of interest.

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
