# Peer review of "Investigation of Vermicompost Influence on Seed Germination of the Endangered Wild Rubber Species Scorzonera tau-saghyz"

_diversity, doi:10.3390/d15020224_

Round 1
Reviewer 1 Report
The work of this manuscript is important and worth publishing. The authors however, need to reach out to an English-speaking scientific colleague that could edit the manuscript writing. There are sections that read as if this would be a student thesis. The materials and methods sections should be direct and concise with little background explanation. For example, the first paragraph of section 2.3 (lines 181-183) should be deleted. It does not belong there. This is a basic rule in scientific journal writing. Additionally, delete all short explanations in parenthesis found throughout the manuscript, for example:
Line 69: …’high polymorphism (there were no cultivars)…
Line 73: …ecosystems, creation gene (seed) bank…
Line 75: …S. tau-saghyz (creation of plantations)….
None of the above are necessary and it is extremely distracting while reading.
Another general comment is Tables and Figures. Table 1 needs to be re-organized in a way that is clear and easy to read it. Figures’ labels need to be concise and/or removed.
This manuscript is not ready for peer-review and much less for publication.
Author Response
Dear Reviewer,
We have eliminated your comment, if there is another comment, we will eliminate.

Reviewer 2 Report
In the manuscript the authors describe the effect of vermicompost/ vermicompost tea on germination and plant growth of endangered species S. tau-saghyz. I believe that the authors collected plant material in compliance with any rules concerning endangered species and stated it in the cover letter. The topic is interesting and important for saving endangered species, unfortunately, the manuscript is not acceptable in present form.
Major comments:
· The authors sometimes do not use appropriate citations – the statement is not mentioned in the text (r. 101 Fabrissin et al., r. 456 Younas et al., …), somewhere the original work is not cited (r. 117 Kavi et al, r. 126 Devi et al…) or popularizing articles without authors in a linked paper are used (for example r. 51 Meer and King, Loo and Mooibroek, ….). Authors should rewrite the Introduction with appropriate scientific citations.
· References, according to Instructions for authors, must be numbered in order of appearance in the text and listed at the end of the manuscript. In the text, reference numbers should be placed in square brackets.
· The aim of the experiment was, as mentioned in point 1: “to investigate the effects of vermicompost tea priming at various concentrations and exposition on germination, seedling vigour of wild-growing Scorzonera tau-saghyz seeds “. Point 2: „to restore the quantity of this endangered species in natural populations” may be mentioned as intention for the future, but it is not solved in this manuscript.
· A more appropriate statistic, not the chi-square test, could have been used to evaluate the results, so that the differences between all variants were visible. How should the control (Figure 3, Table 1) be understood for each variant separately, each with a different result (Table 1)? Statistical results should be presented in tables and graphs, not in text. It's hard to read and confusing.
· Figure 4 is given in the text – r. 273, but it is missing. So, it must be supplemented, or the figures renumbered.
· R. 361-380: This part of the text should be rewritten without repeating the previously mentioned data. Rather, some information may be used in a Conclusion that is too general.
· Somewhere the results are not described completely comprehensibly. I do not understand the statement of 256-257 "...and in 1.32 time less (31.65%) in control group (difference statistically significant at p<0.05), than seed germination rate at 5% VCT priming." How can control be worse than control? What was meant as "the average values" - r. 282, it is the height after 172 days? If so, how can it be smaller than after 28 days (Figures 5 and 6).
· The results must be rewritten to make them clear and concise, see also minor comments.
· The Discussion should be completed. Use more citations and mention the works with similar experiments, for example DOI 10.1007/s10725-011-9586-x
· R. 457-460: The composition of the vermicompost should be described in the methodology
· The last major issue concerns the hard seed coat, which is often mentioned in the manuscript as the main problem with poor seed germination in this species. But hard seed coat is defined as impermeable for water, but in your case, this was probably not the main problem. VKT is not used for the scarification of seeds with a hard coat but acts rather as a nutritional enhancer and contains substances that can disrupt physiological dormancy. You do not mention the presence of substances that would soften the seed coat.
Minor comments:
r. 15: Hevea Brasiliensis
r. 54: did you mean “the number of plants of S. tau-saghyz”?
r. 81: “seedlinging”??? It probably should be seedling growth or seedlings production?
r. 146: How the age of plants was determined?
r. 152-153: Lipschitz
r. 176: “1-3 days”??? There can be big differences between 1- or 3-day maceration.
r. 192: “every 3 days” – What was the total time?
r. 197: “concentration of 20% by weight” Did you mean dry weight or weight with a moisture content of 40%? If you did not calculate it for dry matter (both soil and vermicompost), it would be more appropriate to use volume ratios.
r. 204-205: How was uniform humidity of the substrate ensured?
r. 215: What does "n=3" mean in this case - three pots per variant? If so, that is too little to assess the impact of vermicompost.
r. 295, Fig. 10: Do not use the label "experiment" for variant with vermicompost, control is also a part of this experiment.
r. 331-332: Do not repeat data from graphs or tables.
Reference 37 and citation r.52 Slier et al. should be Siler et al.
Reference 11: Is the name of the chapter correct? Pages are missing.
Reference 41: Rewrite according to instruction for authors and check all references for fullness and correctness.
Author Response
Dear Reviewer,
We have eliminated your comment, if there is another comment, we will eliminate

Round 2
Reviewer 1 Report
Please see my comments and edits in the attached file.
I suggest the manuscript acceptance upon major writing and figures/tables revisions I propose are met.
Best wishes.

Author Response
Dear,
Reviewers, we apologize for sending late, we had objective reasons.

Reviewer 2 Report
A "new comment" has been added to some of the authors' answers:
Point 1: The authors sometimes do not use appropriate citations …….. Authors should rewrite the Introduction with appropriate scientific citations.
New comment: Authors rewrrote the Introduction. But consider grammar (the word order) in the sentence “In our research on the analysis of the quality of tau-sagyz rubber, the method of rubber extraction was used hexane”; and correct “….transferred to the thermostat (temperature + 250C)”…. to 250°C
Point 4: A more appropriate statistic, not the chi-square test, could have been used to evaluate the results, so that the differences between all variants were visible. How should the control (Figure 3, Table 1) be understood for each variant separately, each with a different result (Table 1)? Statistical results should be presented in tables and graphs, not in text. It's hard to read and confusing.
Response 4: We used chi-square test, because it is very simple, and reliable statisticalanalysisi.
It was carry out the series of experiments. In this case for easch experiment were control and experimental data with 1% pre-sowing treatment by VCT. The vigor of germination during 4hours and 8 hours of treatment was not high and we started seed priming with 5% concentration of VCT separetly with control and experimental data for these series of experiment. The results of the short-time treatment by VCT were positive and the strategy of experiments was a correct which confirms the data with 10% VCT treatment.
New comment: If you use a series of experiments, how is it possible to use one Chi-square test for all experiments with a 4-hour and for an 8-hour treatment, both with 3 concentrations + control (in Table 1 you give DF = 1)? Instead of two tests, there should have been six pair-wise tests. You probably added up all the controls and all the VCT concentrations. But this is not a good way because different concentrations had a different effect. For example, there is a big difference between control and 10% VCT in 4-hours treatment, but no difference using 1% VCT. In addition, when using the chi-square test, we do not see differences between individual concentrations. Another statistical test could have been used to compare the results completely.
Point 5: Figure 4 is given in the text – r. 273, but it is missing. So, it must be supplemented, or the figures renumbered.
Response 5: - The figures will renumbered beginning with Figure 4 in correct order
New comment: Figure 4 is still missing.
Point 6: R. 361-380: This part of the text should be rewritten without repeating the previously mentioned data. Rather, some information may be used in a Conclusion that is too general.
Response 6: The part of text (p. 362-372) will include in p.443. The other part of text (p. 373-376) will insert in p.489, the text( p.377-381) will insert in p. 492.
New comment: I do not see these changes in the text.
Point 7: Somewhere the results are not described completely comprehensibly. I do not understand the statement of 256-257 "...and in 1.32 time less (31.65%) in control group (difference statistically significant at p<0.05), than seed germination rate at 5% VCT priming." How can control be worse than control? What was meant as "the average values" - r. 282, it is the height after 172 days? If so, how can it be smaller than after 28 days (Figures 5 and 6).
Response 7: Please provide your response for Point 7. (in red)
New comment: Response is missing. The results must be rewritten to make them more clear.
Point 8:The results must be rewritten to make them clear and concise, see also minor comments.
Response 8: Please provide your response for Point 8. (in red)
New comment: Response is missing.
Point 9:R. 457-460: The composition of the vermicompost should be described in the methodology
Response 9: The composition of the vermicompost will transferred in the Chapter "Methodology" in p.170
New comment: The change was not done.
Point 11: r. 15: Hevea
Response 11: (Hevea brasiliensis)
New comment: Should be written in Italics.
Point 17: r. 192: “every 3 days” – What was the total time?
Response 17: (within 2 months.)
New comment: Information should be add to the text.
Author Response

(The authors gave the same response as above.)

Round 3
Reviewer 1 Report
The manuscript is much more improved and is now ready for publication. I accept the new version in the present form.